# Pulmonary acini exhibit complex changes during postnatal rat lung development

**David Haberthür** [iD]**, Eveline Yao, Sébastien F. Barré, Tiziana P. Cremona, Stefan A. Tschanz, Johannes C. Schittny** [iD]*****

Institute of Anatomy, University of Bern, Bern, Switzerland

* schittny@ana.unibe.ch

**Data Availability Statement:** The data underlying this study can be accessed at https://doi.org/10.17605/OSF.IO/JKUNE.

**Funding:** We are thankful for the support by the Swiss National Science Foundation (grants

## Abstract

Pulmonary acini represent the functional gas-exchanging units of the lung. Due to technical limitations, individual acini cannot be identified on microscopic lung sections. To overcome these limitations, we imaged the right lower lobes of instillation-fixed rat lungs from postnatal days P4, P10, P21, and P60 at the TOMCAT beamline of the Swiss Light Source synchrotron facility at a voxel size of 1.48 μm. Individual acini were segmented from the three-dimensional data by closing the airways at the transition from conducting to gas exchanging airways. For a subset of acini (N = 268), we followed the acinar development by stereologically assessing their volume and their number of alveoli. We found that the mean volume of the acini increases 23 times during the observed time-frame. The coefficients of variation dropped from 1.26 to 0.49 and the difference between the mean volumes of the fraction of the 20% smallest to the 20% largest acini decreased from a factor of 27.26 (day 4) to a factor of 4.07 (day 60), i.e. shows a smaller dispersion at later time points. The acinar volumes show a large variation early in lung development and homogenize during maturation of the lung by reducing their size distribution by a factor of 7 until adulthood. The homogenization of the acinar sizes hints at an optimization of the gas-exchange region in the lungs of adult animals and that acini of different size are not evenly distributed in the lungs. This likely leads to more homogeneous ventilation at later stages in lung development.

## Introduction

### Lung development

Lung development starts with the formation of the two lung buds. During the embryonic stage (prenatal days E11–E13 in rats) the major airways and the pleura are formed. During the pseudoglandular stage (E13–E18.5) most of the remaining airway generations are formed by branching morphogenesis. Only a small amount of additional branches are added during the later stages (canalicular: E18.5–E20) and saccular: E20 to postnatal day P4) [1]. Epithelial differentiation during the canalicular stage and angiogenetic activation of the capillaries lead to the first functional air-blood barriers in the lung [2]. The saccular stage represents an intermediate stage where branching morphogenesis ceases and the developmental program switches to the phase of alveolarization. During alveolarization, 90% of the adult gas-exchange area is

310030_153468 and 310030_175953). The funders had no role in study design, data collection and analysis, decision to publish, or preparation of the manuscript.

**Competing interests:** No conflicts of interest, financial or otherwise, are declared by the authors.

formed by lifting off new septa from the existing gas-exchange surface [3–5]. Alveolarization itself is divided into two distinct phases, the so-called classical (P4–P21) and continued alveolarization (P14 to approximately P60) [6, 7]. During classical alveolarization, new septa are formed starting from preexisting immature septa which contain a double-layered capillary network. During continued alveolarization, new septa are formed from preexisting mature septa. In general, during alveolarization the structure of the alveolar septa is changed to increase the efficiency of the lung [6, 7].

In this manuscript, the timing of lung development is given for rats while the same stages and developmental mechanisms are observed in every placentalian species studied so far, including humans. However, the timing of lung development and especially the time point of birth relative to lung development is adapted in each species [2, 5, 7].

## The functional lung units

Pulmonary acini represent the gas-exchanging units of the bronchial tree and are defined as the airways distal of the terminal bronchioles [8]. In humans, the acini contain approximately four generations of respiratory bronchioles before reaching the alveolar ducts. The alveolar ducts start at the so-called bronchioalveolar duct junction (BADJ) where the lining of the inner airway surface abruptly changes from the cuboidal epithelium of the bronchioles into type 1 and type 2 alveolar epithelium which covers the alveoli. The small tree of airways distal of the BADJ is called a ventilatory unit [9]. Since murine lungs do not possess respiratory bronchioles, the murine acini consist of one single ventilatory unit [9]. In this study we used rat lungs; hence we exclusively speak of acini. To translate our results from rats to humans, monkeys, or dogs one would have to compare one rat acinus to one single ventilatory unit in these species.

Developmentally, the BADJ is of particular interest. The junction is formed during the canalicular stage when the epithelia are differentiating. The generation of the airway in which an individual murine acinus starts is defined during the canalicular stage and remains constant once it is formed. As a consequence, in rats the number of formed acini also remains constant during the phase of alveolarization and thereafter [10, 11]. The latter was somehow surprising because in rats the total lung volume increases by roughly a factor of 10 during alveolarization [6]. Therefore, the mean volume of the acini has to increase by approximately the same factor.

The acinar architecture is important for ventilation, airflow [12–14] and particle deposition [6, 15, 16].

## Acinus detection and delineation

On single two-dimensional sections—either histological or virtual from tomographic imaging—it is not possible to unambiguously detect which parts of the airways and alveoli are connected three-dimensionally. To detect and delineate individual acini the full three-dimensional information is needed. Historically, only a small amount of acini were delineated and extracted from manually traced serial sections [17, 18] or sectioned silicone casts [19] with considerable manual work. Non-destructive three-dimensional imaging is best suited to acquire data sets that can be used to easily detect, delineate, and study large amounts of single acini. This has been shown by Vasilescu et al. [20]. In their study, they reconstructed 22 mouse acini from tomographic data sets of four 12 weeks old mice (at least postnatal day 84, fully developed lungs) and compared those with scanning electron microscope images of acinar silicone rubber casts. Haberthür et al. [21] analyzed the volume, surface area, and number of alveoli of 43 individual acini of three adult rats (day 60) in tomographic datasets obtained by synchrotron radiation based X-ray tomographic microscopy. Similarly, Kizhakke Puliyakote et al. [22]

analyzed the volume of 32 mouse acini from six animals, as well as the branching pattern of their internal airways. All prior work known to us has analyzed considerably fewer acini than the 268 acini presented here and focuses only on a time point, at which the lungs of the animals are already fully developed while our data spans the postnatal days 4 to 60.

### Aims of this study

**Acinar structure.**   Due to the lack of precise and complete three-dimensional data, simulations of airflow in the lung are based on simple idealized acinar models [23, 24]. The question of how well these models represent lung physiology remains open until the necessary data and physiologically correct models are available. To contribute to the validation of the computational fluid dynamics simulations, we extracted individual acini and determined their volume and their number of alveoli throughout rat lung development.

**Individual acinar volumes.**   While the global volume of the acini is well known and studied, only little is known about the range of the volume of *individual* acini throughout lung development. The method we developed is suited to analyze large amounts of individual acini over the course of postnatal lung development in a time-efficient manner. Data on such large amounts of acini was not available up to now.

Our aim was to understand the size distribution of the acini during lung development including the contribution of the rather irregular shape of the alveoli. The observed range *de*creased during postnatal lung development by a factor of 6–7, which represents an unexpected result. The obtained data is essential for further investigations, like the influence of the acinar size on ventilation and gas-washout, or the influence of the irregular alveolar shape on particle deposition inside individual acini. Ongoing work (unpublished results and [13]) is using a computational fluid dynamics model to simulate exactly this influence. In particular, the washout time increases with increasing acinar size, acinar size distribution, and compliance.

## Materials & methods

### Rat lung samples

In the present study, a superset of the animals described by Haberthür et al. [21] were used. Tschanz et al. [6] described a biphasic formation of new alveoli on the same set of animals. The stereological analysis presented here represents a part of the 3R-initiative (replacement, reduction, and refinement) [25] for the ethical use and the reduction of the number of animals sacrificed for science.

Briefly, we extracted the lungs of Sprague-Dawley rats after having deeply anesthetized them with a mixture of medetomidine, midazolam, and fentanyl [26]. The rats were euthanized by exsanguination during the removal of the lung. The lungs were fixed with 2.5% glutaraldehyde by intratracheal instillation and kept under constant pressure (20 cm $H_2O$) during fixation to prevent recoiling of the tissue. The samples were prepared for tomographic imaging by post-fixation with 1% osmium tetroxide and staining with 4% uranyl acetate, as well as by dehydration in a graded series of ethanol and embedding in paraffin. In total we assessed 12 animals, on postnatal days P4, P10, P21, and P60 (N = 3 per day. From now on, we will omit the postnatal prefix and only mention the day.).

The animals were housed in the central animal facility of the University of Bern. They received food and water ad libitum at a 12/12 hours day/night cycle. The experiments themselves, as well as the handling of the animals before and during the experiments, were approved and supervised by the Federal Food Safety and Veterinary Office of Switzerland and the Department of Agriculture and Nature of the Kanton of Bern in Switzerland.

## Tomographic imaging

Tomographic imaging was performed at the TOMCAT beamline (A beamline for TOmographic Microscopy and Coherent rAdiology experimenTs) [27, 28] of the Swiss Light Source at the Paul Scherrer Institute in Villigen, Switzerland. The samples were scanned at an X-ray energy of 20.0 keV. After penetrating the sample, the X-rays were converted into visible light by either a 20 μm-thick LuAG:Ce or 18 μm-thick YAG:Ce scintillator screen (both from Crytur, Turnov, Czech Republic), depending on the date of experiments. The resulting visible light was magnified using a 10-times magnifying, diffraction-limited microscope lens and recorded with a 2048 × 2048 pixel CCD camera (pco.2000, PCO, Kelheim, Germany) with 14 bits dynamic range operated in 2 by 2 binning mode. As a result, in a single field of view, we were able to image a sample volume of a cube of 1.5 mm side length with a voxel side length of 1.48 μm, with the exposure time of the single projections varying between 160 and 200 ms.

Since our samples were larger than the field of view of the detector, we applied the so-called wide-field scanning method [29] to increase the field of view horizontally. For each sub-scan, we recorded 3578 projections and laterally stitched them so that their combined field of view covered the whole sample width. Additionally, two or three such wide-field scans were stacked vertically to match the sample height.

The resulting data sets for each of the lung samples covered a field of view of a cube with approximately 4.5 mm side length which corresponds to an image stack with approximately 3000 × 3000 × 3000 pixels at 1.48 μm side length each. The mean size of the tomographic reconstructions, corresponding to the raw data of each of the 42 analyzed tomographic scans is 8 GB, totaling to 336 GB.

## Extraction of acini

For the present manuscript, we extracted single acini from three-dimensional data, acquired without destroying the samples. Details of this semi-automatic extraction of the rat acini are described by Haberthür et al. [21]. Individual acini were extracted from the tomographic data with a custom image processing network established in MeVisLab (version 2.1, 2010-07-26 release, MeVis Medical Solutions and Fraunhofer MEVIS-Institute for Medical Image Computing, Bremen, Germany). To isolate individual acini, we followed the airway tree in the data and manually placed disk-shaped segmentation stoppers at the acinus entrances. Thus, acini were selected independently of their size and only based on their location in the airway tree. The acinus extraction was performed by segmenting the acinus volume with a gray-level threshold-based region growing algorithm with a seed point perpendicular to the manually placed segmentation stopper. Each individual acinus was then exported as a single DICOM file for portability. For each acinus, its exported DICOM file contained the segmented acinus overlaid with the original background for further analysis. Example images from the DICOM files are shown in panels I–L of Fig 1.

This semi-automatic acinus segmentation method was applied to lung samples obtained throughout lung development, at postnatal days 4, 10, 21 and 60. In total we extracted 701 acini from 12 animals, a breakdown of the number of extracted and assessed acini per animal is given in Table 1. Even if our method is efficient, we could not have stereologically assessed all these 701 acini, we thus performed a subsampling and stereologically analyzed 268 acini, as specified in the section below.

## Stereological analysis of alveolar characteristics

The stereological estimation of the alveolar number was performed according to the standards for quantitative assessment of lung structure from the American Thoracic Society [30] to guarantee accurate and unbiased results. As a basis for the estimation the volumes of the lung lobes

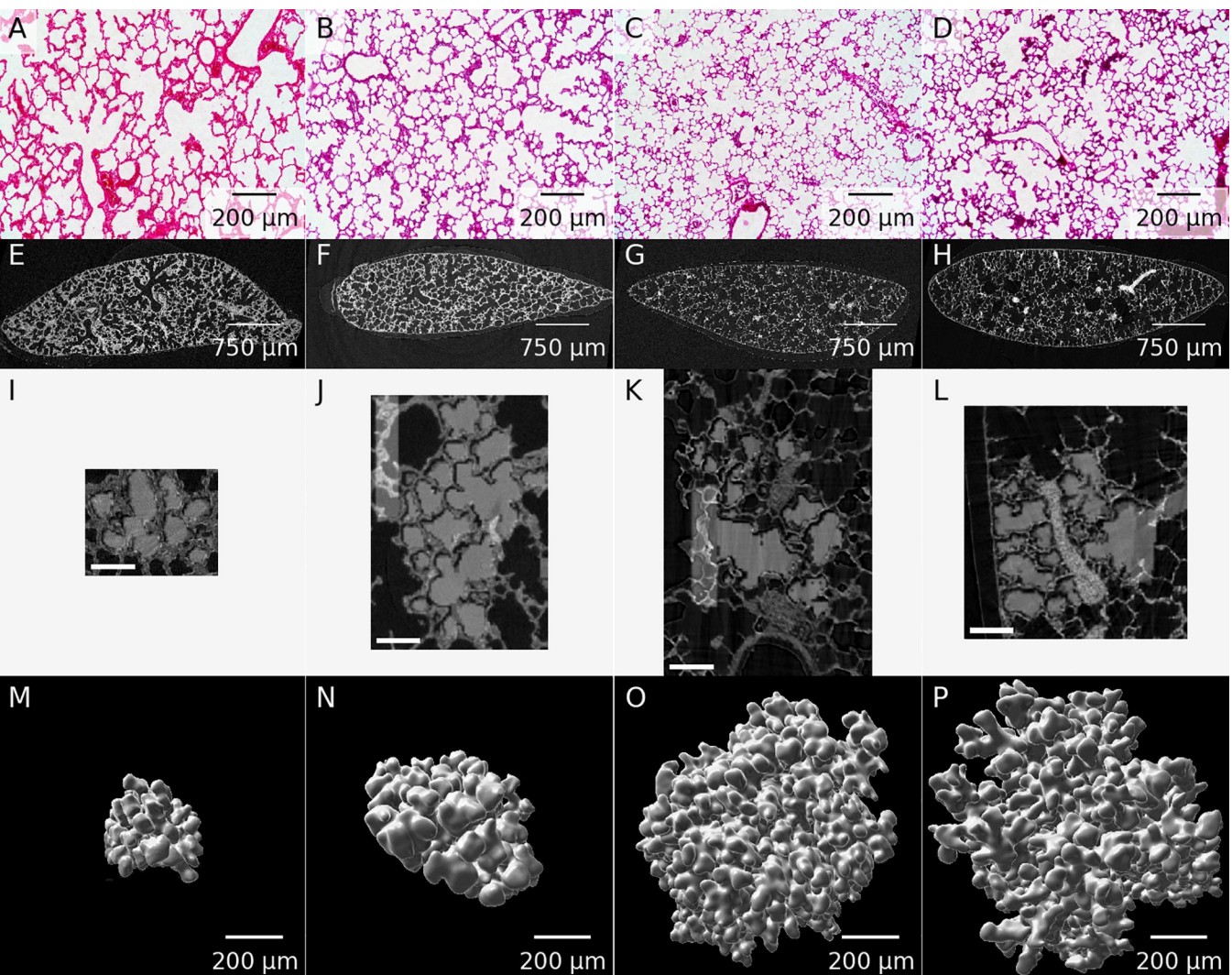

**Fig 1. Example images for each of the assessed time points.** A–D: Representative light microscopy images. E–H: One slice of the tomographic data sets. I–L: Middle slice of one of the extracted acini. The extracted acinus is overlaid on the tomographic dataset. Panels J and K also show the segmentation stopper at the left border of the image. The shown images correspond to the middle slice of the full data sets used for the stereological analysis. The scale bar was burnt in during preparation of the image stacks and is 100 µm long. M–P: Three-dimensional visualization of one of the extracted acini. Panels A, E, I & M: Day 4. Panels B, F, J & N: Day 10. Panels C, G, K & O: Day 21. Panels D, H, L & P: Day 60.

were first determined by water displacement [31]. The lobe-volume was estimated a second time by the Cavalieri method. Both volumes were used to calculate the shrinkage, which was used for the correction of the volume data [6].

To perform the stereological analysis, each DICOM file from the isolated acini was sampled into a systematic random set of single images using a MATLAB script [32]. The stereological assessment was performed with the STEPanizer [33], an easy-to-use software tool for the stereological assessment of digital images. Details of the stereological assessment have previously been described [21]. Briefly, we counted the appearance or disappearance of alveolar wall interruptions. Under the assumption that these only occur in the region of the alveolar mouth opening and correspond to alveolar entrance rings [34] we counted said interruptions on paired images spanning the whole acinus volume. By using the disector method [35] we were thus able to estimate the number of alveoli in 268 of the 701 exported acini. The acini we

analyzed were subsampled from the full data by systematic uniform random sampling [30]. A defined fraction (i.e. one third) of all the extracted acini of one animal were analyzed. During segmentation, the acini were numbered. Based on these numbers every third acinus was selected, randomly starting with the first, second, or third one.

Previously, we have shown that the stereological assessment of the acinar volume (according to the Cavalieri principle [36]) gives comparable results to an automated volume extraction by counting the segmented pixels in the tomographic data sets [21]. Due to variations in the gray value, the automatic segmentation underestimates the volume of the single acinus. The best-suited approach to obtain unbiased results is to assess the volume manually according to the Cavalieri principle, which is what we did for the 268 acini presented in this manuscript. The sets of JPG slices and the raw results from the stereological analysis with the STEPanizer are available online [37].

### Data analysis and display

All the stereologically assessed data was processed in a Jupyter [38] notebook, producing all the results, including the statistical data and plots shown below. The notebook with all its history is freely available on GitHub [39]. The performed calculations are described in detail in the Results section. In the plots, semitransparent circles mark the single observations. The box shows the 25%–75% quartiles range of the data. The whiskers extend to the rest of the distribution. Outliers are determined as a function of the inter-quartile range and are shown outside the whiskers. Numerical values in the text are given as averages ± standard deviation. P-values in the text and figure legends are given as precise numbers, as suggested by Amrhein et al. [40].

Usually, we performed a Shapiro-Wilk test for normality [41] to test whether we can use an U-test for assessing the differences, namely a two-sided Mann-Whitney rank test [42]. This rank test was used to assess the differences between the possible combinations. An additional Kruskal-Wallis H-test for independent samples [43] was used to test for sample independence. The statistical analysis was performed in the aforementioned notebook by using the statistical functions of the Scientific Computing Tools for Python [44] or GraphPad Prism 7.01 (Graph-Pad Software, San Diego, CA, USA).

## Results

The number of alveoli per acinus was assessed for 268 individual acini at days 4, 10, 21 and 60. For day 4, we analyzed 125 acini, for day 10 we analyzed 58 acini, for day 21 we analyzed 42 acini, and for day 60 we analyzed 43 acini.

An overview of our process is shown in Fig 1. Panels A–D depict a representative light microscopy image for each assessed time point. Panels E–H of Fig 1 each correspond to one slice of the tomographic data sets acquired at TOMCAT. Panels I–L of Fig 1 show the middle slice of the datasets used for the stereological analysis of the extracted acini. The acinus is shown in light gray as an overlay over the tomographic data. Panels M–P of Fig 1 show three-dimensional visualizations of representative acini.

### Alveoli per acinus

The average entrance ring count, which corresponds to the number of alveoli per acinus for the 125 acini at day 4 is 48 ± 41 alveoli. For the 58 acini at day 10 it is 89 ± 84 alveoli, for the 42 acini at day 21 it is 233 ± 164 alveoli. At day 60 we assessed 43 acini in total, on average they have 702 ± 219 alveoli.

The values for the single animal are given in Table 1 and plotted in Fig 2.

**Table 1. Detailed alveolar numbers for each animal.**

| Animal | Assessed acini | Average counts | STD | Minimum | Maximum |
|---|---|---|---|---|---|
| 04A | 51 | 26.73 | 23.56 | 4 | 111 |
| 04B | 23 | 65.04 | 43.85 | 15 | 171 |
| 04C | 51 | 60.51 | 45.42 | 10 | 249 |
| 10A | 27 | 77.85 | 56.31 | 18 | 245 |
| 10B | 14 | 84.57 | 65.75 | 23 | 199 |
| 10C | 17 | 108.76 | 125.45 | 15 | 505 |
| 21B | 14 | 208.14 | 197.71 | 35 | 781 |
| 21D | 17 | 196.00 | 128.56 | 50 | 493 |
| 21E | 11 | 323.09 | 145.94 | 108 | 572 |
| 60B | 24 | 701.75 | 230.95 | 322 | 1296 |
| 60D | 10 | 668.30 | 121.67 | 444 | 810 |
| 60E | 9 | 739.11 | 281.22 | 272 | 1204 |

The alveolar number shows highly significant differences (Šidák-corrected p-value smaller than 0.00167 [45]) between all possible combination of days. All possible combinations of entrance ring counts per day are significantly different (all p-values are smaller than p = 1.9e-5, which is the p-value of the difference between days 4 and 10). The entrance ring counts for all animals are independent (p = 1.6e-34).

## Acinus volume

The stereological assessment resulted in a mean acinar volume of $0.03 \pm 0.04$ mm$^3$ for the 125 acini at day 4. For the 58 acini at day 10 we get a volume of $0.04 \pm 0.05$ mm$^3$. For the 42 acini

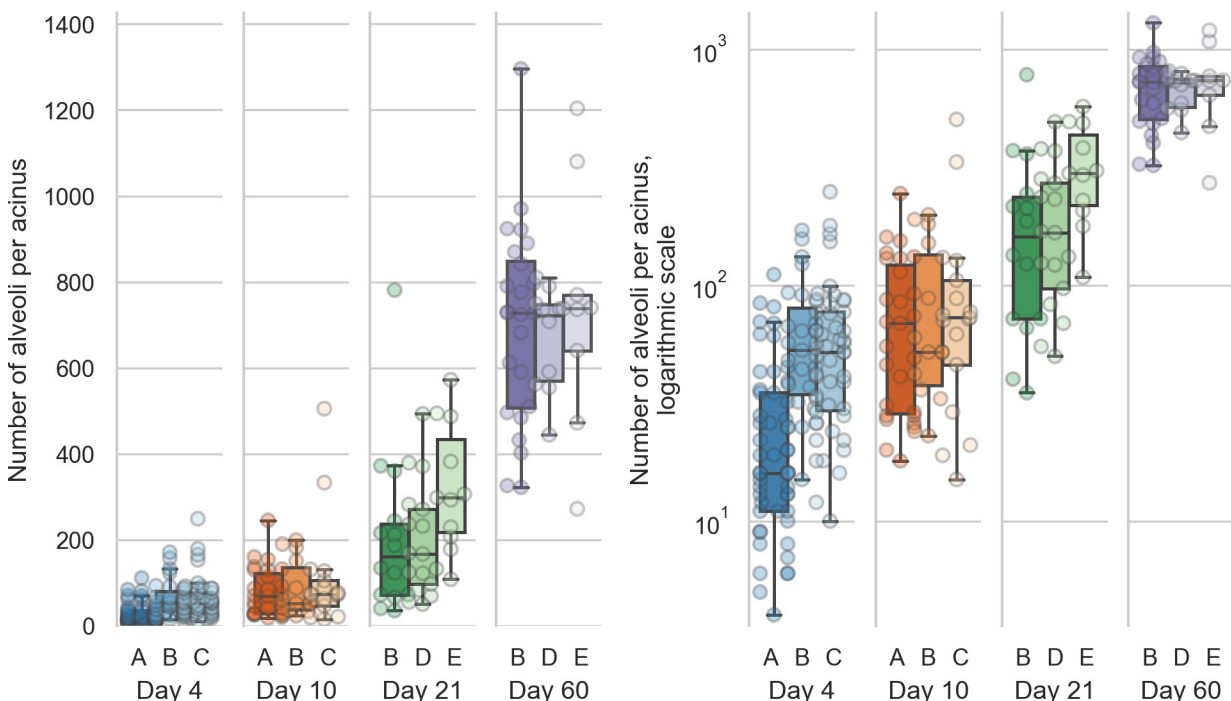

**Fig 2. Distribution of the number of alveoli per acinus (entrance ring count per acinus) for each animal.** Left: linear scale, right: logarithmic scale. The entrance ring counts per day are all significantly different (all p-values smaller than 1.9e-5, which is the p-value between days 4 and 10).

**Table 2. Detailed volume data (in mm³) for each animal.**

| Animal | Assessed acini | Average [mm³] | STD [mm³] | Minimum [mm³] | Maximum [mm³] |
|---|---|---|---|---|---|
| 04A | 51 | 0.009 | 0.008 | 0.002 | 0.035 |
| 04B | 23 | 0.058 | 0.053 | 0.008 | 0.194 |
| 04C | 51 | 0.044 | 0.044 | 0.006 | 0.275 |
| 10A | 27 | 0.033 | 0.026 | 0.004 | 0.108 |
| 10B | 14 | 0.033 | 0.030 | 0.004 | 0.096 |
| 10C | 17 | 0.057 | 0.084 | 0.002 | 0.333 |
| 21B | 14 | 0.087 | 0.086 | 0.011 | 0.322 |
| 21D | 17 | 0.109 | 0.112 | 0.014 | 0.380 |
| 21E | 11 | 0.118 | 0.072 | 0.025 | 0.238 |
| 60B | 24 | 0.588 | 0.223 | 0.226 | 1.104 |
| 60D | 10 | 0.888 | 0.350 | 0.364 | 1.576 |
| 60E | 9 | 0.977 | 0.526 | 0.186 | 1.875 |

at day 21 a volume of $0.10 \pm 0.09$ mm³ and for the 43 acini at day 60 a volume of $0.74 \pm 0.37$ mm³.

The values for the single animals are shown in Table 2 and displayed in Fig 3.

The mean volume of the 20% smallest acini to the 20% largest acini at day 4 differs by a factor of 27.47 (from 0.0035 μl–0.095 μl, N = 25). At day 10, this increase is 15.28 times (from 0.0071 μl–0.11 μl, N = 12). At day 21, this increase is 14.42 times (from 0.018 μl–0.26 μl, N = 8). At day 60, we saw an increase of 3.94 times (from 0.33 μl–1.3 μl, N = 9).

The acinar volumes show highly significant differences between all possible combinations of days except between days 4 and 10 (p = 0.08). All other p-values are smaller than 4.5e-6, which is the p-value for the difference between days 10 and 21. The statistical analysis is equal to what is briefly described for the entrance ring count above. The acinar volumes for animals 21E and all animals of day 60 are non-normally distributed, but all acinar volumes per animal are independent (p = 5e-29).

To assess the volume distribution per day, we normalized the volumes per day to the largest volume per day (Fig 4). The whiskers in the plots contain all data points 1.5 times the interquartile range past tow low and high quartiles, points outside the whiskers are considered outliers. The non-outlier data points are closest together on day 4 and slightly more spread out at day 10. At days 21 and 60, we see no outliers for the normalized acinar volumes. At days 4, 10 and 21 the distribution of the normalized volumes is skewed towards below the average (median of normalized volumes at day 4: 0.16, at day 10: 0.18 and at day 21: 0.23), we thus see more smaller than larger acini early in the development. At day 60, the distribution is more homogeneous, with the median of the normalized acinus volumes at exactly 0.50.

## Alveoli per volume

The mean number of alveoli for the 125 acini at day 4 is $2166 \pm 1132$ alveoli mm⁻³. For the 58 acini at day 10 we counted $2831 \pm 1189$ alveoli mm⁻³, for the 42 acini at day 21 $2723 \pm 771$ alveoli mm⁻³ and for the 43 acini at day 60 $1080 \pm 349$ alveoli mm⁻³. These numbers were found by dividing the counted entrance rings by the acinus volumes.

The numbers for the single animals are shown in Table 3 and displayed in Fig 5.

The counts per volume—which correspond to the number of alveoli per acinus volume—are one measures for the complexity of the single acini. The number of alveoli per acinus show highly significant differences between all possible combinations of days except between days 10 and 21 (p = 0.7). All other days have a p-value smaller than 5e-5, this being the p-value

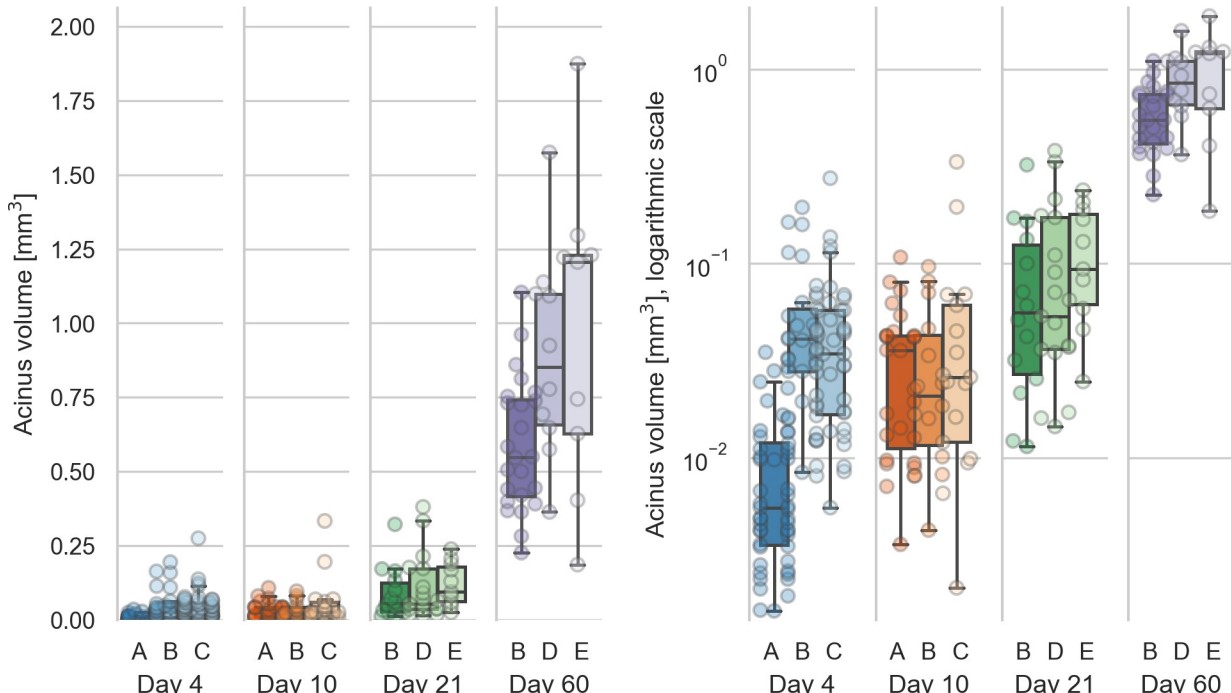

**Fig 3. Distribution of the acinar volumes for each animal.** Left: linear scale, right: logarithmic scale. The acinar volumes are all significantly different (all p-values smaller than 4.5e-6, which is the p-value between days 10 and 21) for each combination of days except between days 4 and 10 (p = 0.08).

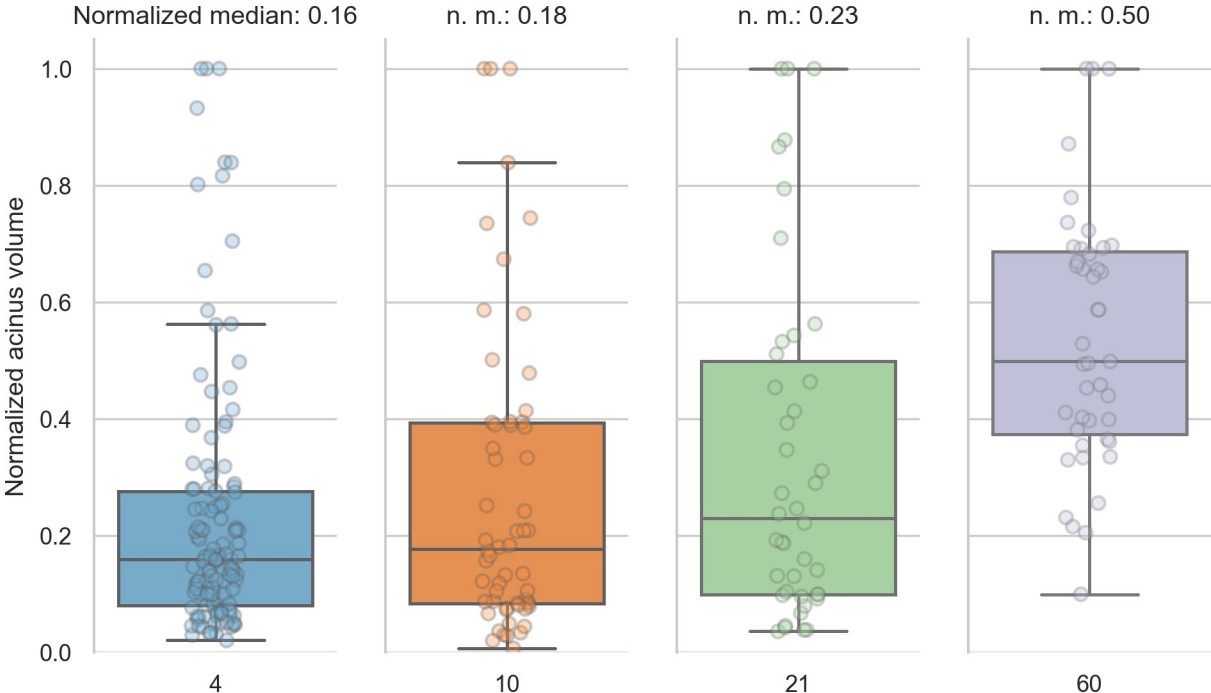

**Fig 4. Normalized acinar volumes.** The spread of the volumes is small early in the development and skewed towards the lower end of the volumes. At day 60, the median is exactly in the middle of the normalized volumes.

**Table 3. Details of alveolar density (number per mm³) per acinus volume for each animal.**

| Animal | Assessed acini | Average [mm⁻³] | STD [mm⁻³] | Minimum [mm⁻³] | Maximum [mm⁻³] |
|--------|----------------|----------------|------------|----------------|----------------|
| 04A | 51 | 3124 | 1152 | 1618 | 8208 |
| 04B | 23 | 1363 | 499 | 679 | 2704 |
| 04C | 51 | 1569 | 380 | 799 | 2775 |
| 10A | 27 | 2685 | 1022 | 1246 | 5582 |
| 10B | 14 | 3091 | 1166 | 1890 | 5422 |
| 10C | 17 | 2848 | 1462 | 1182 | 7009 |
| 21B | 14 | 2642 | 493 | 1625 | 3344 |
| 21D | 17 | 2534 | 895 | 1111 | 4032 |
| 21E | 11 | 3119 | 775 | 1604 | 4379 |
| 60B | 24 | 1249 | 297 | 530 | 1912 |
| 60D | 10 | 838 | 271 | 476 | 1221 |
| 60E | 9 | 898 | 323 | 411 | 1463 |

between days 4 and 21. The statistical analysis is equal to what is briefly described for the entrance ring count above. The counts per volume for all animals except 04A are non-normally distributed, but all are independent (p = 2.9e-21).

## Number of acini

The number of acini for day 4 was 18260 ± 16555 acini for day 10 14359 ± 5611 acini, for day 21 11203 ± 2681 acini and for day 60 4277 ± 777 acini. These numbers were obtained by

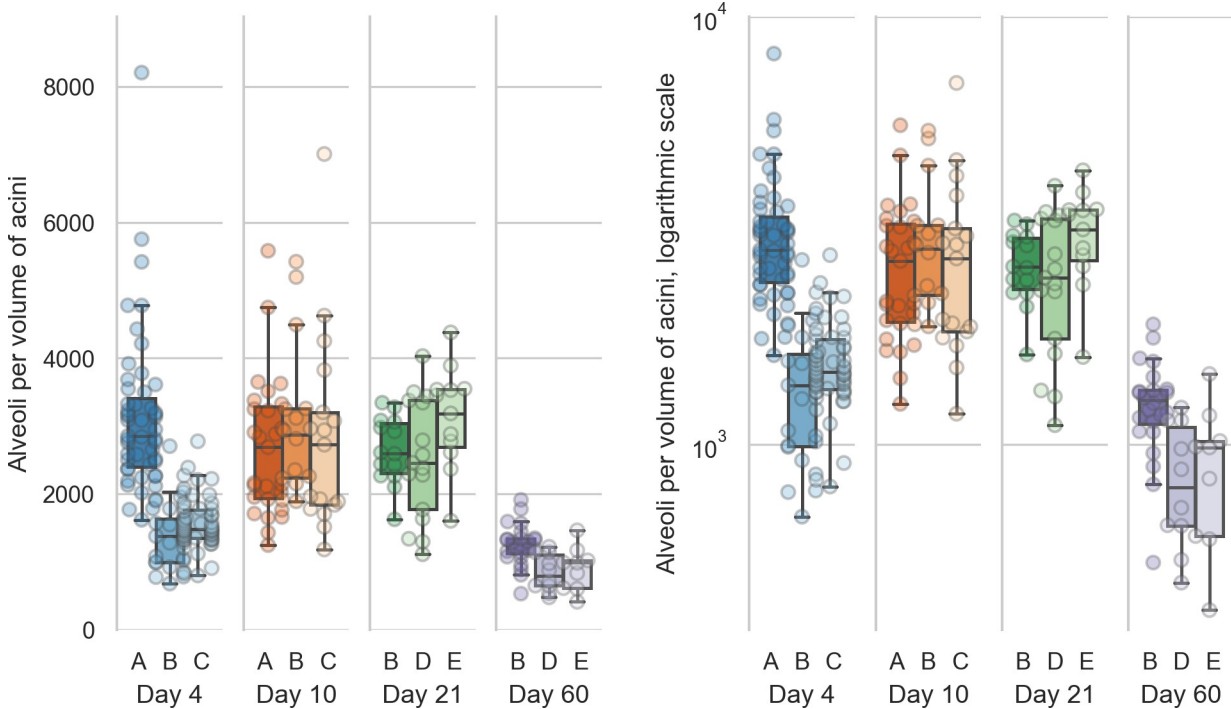

**Fig 5. Number of alveoli per acinus volume.** Left: linear scale, right: logarithmic scale. The number of alveoli per acinus volume are all significantly different (all p-values are smaller than 5e-5, which is the p-value between days 4 and 21) for each combination of days except between days 10 and 21 (p = 0.7).

dividing the parenchymal volume of the lungs [6] with the Cavalieri-estimated volume of the acini. To calculate these data we had to use data of two different studies.

## Number of alveoli per lung

The total alveolar number for day 4 was found to be 0.64 ± 0.34 million alveoli, for day 10 1.23 ± 0.39 million alveoli, for day 21 2.65 ± 0.57 million alveoli, and for day 60 3.01 ± 0.34 million alveoli. The total number of acini was estimated by dividing the mean parenchymal volume of the lungs [6] by the stereologically estimated acinar volume. The number of alveoli for each animal is then calculated by multiplying the average entrance ring count per animal with the estimated number of acini per animal. To calculate this data we had to use data from three different studies performed by three different teams using two different strains of animals (i.e. current data, Tschanz et al. [6], and Barré et al. [11]). Therefore, we had to accept that error propagation may cause a significant blurring of our results.

## Volume of individual alveoli

The average volume of one alveolus including its part of the alveolar duct was estimated by dividing the mean acinar volume by the mean count of alveoli for each day separately, while correcting for the ductal volume. For day 4 we estimate the volume of an individual alveolus at $4.98 \times 10^5$ μm$^3$, for day 10 at $3.34 \times 10^5$ μm$^3$, for day 21 at $3.07 \times 10^5$ μm$^3$ and for day 60 at $8.12 \times 10^5$ μm$^3$. This corresponds to an average diameter per alveolus of 98 μm, 86 μm, 84 μm and 116 μm for days 4, 10, 21 and 60, respectively. The average diameter was estimated by assuming spherical alveoli and solving the volume equation of a sphere to its diameter.

## Discussion

To our best knowledge, the present manuscript is the first to quantitatively describe the number of alveoli in individual acini and changes in the volumes of single rat lung acini. Based on stereological analysis of a large number of individual acini we were able to efficiently extract qualitative and quantitative data from non-destructive three-dimensional tomographic data sets of individual acini in a highly reproducible manner.

We can easily estimate the total number of acini during lung development from our data. The limitations of this study are mainly twofold: a) due to technical reasons (i.e. available beamtime and limitations of the setup) we were not able to image the entire lung and b) lung location (i.e. lower medial tip of the right lobe). However, our sample is more than suitable for the drawn conclusions because it was previously shown that the right lower lobe is a valid sample for the entire lung [10] and that information drawn from one lobe is representative of the whole lung [46]. Furthermore, based on stereological data, Zeltner et al. [46] were able to show that the lung parenchyma is rather homogeneous. Looking at individual acini, Kizhakke Puliyakote et al. [22] observed that the peripheral, pleura facing acini possess an acinar volume which is two thirds larger than the volume of the central ones. However, they did not report additional regional differences, for example between acini located cranially and caudally. Own preliminary data point to the same result. We imaged an entire rat lung *in situ* immediately post mortem using high-resolution synchrotron radiation based X-ray tomographic microscopy [47]. A first analysis of regional difference shows similar results as mentioned above.

## Acinus volume

The mean acinar volume significantly increases during lung development from postnatal days 10 to 60 with all p-values smaller than 4.5e-6. No significant difference was observed between

postnatal days 4 and 10 (1.2× increase, p = 0.08). A large part of the acinus volume increase happens after day 21 (compare Fig 3), which is consistent with previous literature [3, 6].

As mentioned in the introduction section, Barré et al. [10] have shown that the number of acini remains constant during lung development, albeit in a different strain of animals (Wistar Bern compared to Sprague-Dawley in this study). The total volume of the alveolar space increases by roughly a factor of 11 during alveolarization [6]. Therefore, we expect the mean volume of the acini to increase by at least the same factor.

We observed a mean increase of the acinus volumes of 22.66 times between days 4 and 60 (p = 1.8e-22). The other combinations of increases were: day 4 to day 21: 3.18× (p = 1.3e-9), day 10 to day 21: 2.60× (p = 4.5e-6), day 10 to day 60: 18.46× (p = 1.4e-17), day 21 to day 60: 7.11×(p = 7.1e-15). The increase from day 4 to day 10 was not significant (p = 0.08) at 1.23×).

During the same time, the coefficients of variation of the acinus volumes dropped from 1.26 to 0.49. At day 4, the mean volumes of the 20% largest acini are 27.47 times larger than the volumes of the 20% smallest acini (N = 25). At day 60, the 20% largest acini are 3.94 times larger than the 20% smallest acini (N = 9). The size distribution of the largest to smallest acini thus decreases nearly 7 times over the course of the lung development. This indicates that the dispersion of the value is based on an entirely different pattern of distribution of acinar sizes and that the difference of the mean acinus volumes from the smallest to the largest fraction per day becomes smaller and smaller during postnatal lung development.

When comparing the acinus volumes normalized to the largest volume per day, we found that the volumes are closer together on day 4 and are spread out more on the other days (Fig 4). The normalized volumes (without outliers) are most compact on day 4 and slightly more spread out at day 10. At days 21 and 60, we see no outliers for the normalized acinar volumes, which thus range from zero to one. The median of the normalized volumes is smaller than the average at days 4, 10 and 21 while it is exactly at 0.5 for day 60. This means that we have more smaller than larger acini early in the development and a more homogeneous distribution late in lung development. This result suggests that acinar growth is not constant during lung development. It seems that some acini increase their volume faster than others. As a result, we speculate that an increased percentage of acini is somehow dormant at day 4-21, while at day 60 most of the acini reached their final volume.

By dividing the parenchymal volumes of each lung estimated by Tschanz et al. (Table 1, [6]) by the mean number of acini estimated by Barré et al. [11] we can estimate the mean acinar volume for each day. The acinus volume we estimated by point counting (Cavalieri estimation) are on average 2.07 times lower but show a similar increase over the studied period (see Fig 6). It appears that the comparison between data obtained by stereology based on classical paraffin sections or on any kind of 3D-imaging has its limitation if it comes to the absolute numbers. First, due to the *coast of Britain*-effect the exact value of every surface measurement is depending on the resolution of the imaging method [48]. Second, the thickness of the alveolar septa in 3D-imaging data is heavily depending on the segmentation between airspace and tissue—regardless if it is done manually or by a computed algorithm. We observed that the airspace volumes obtained by 3D-imaging has a tendency to be smaller than the ones determined using classical paraffin sections. Nonetheless, the relative numbers between studies to match well.

## Number of alveoli per acinus volume

The volume and shape of the alveoli, alveolar ducts, and the acini are critical parameters for ventilation and particle deposition [49–51]. Particle deposition, for example, is low during the first two weeks of a rats' life, high at postnatal day P21 and medium at days P36–P90 [15, 16, 52]. This correlates well with our results, where we found a significantly larger number of

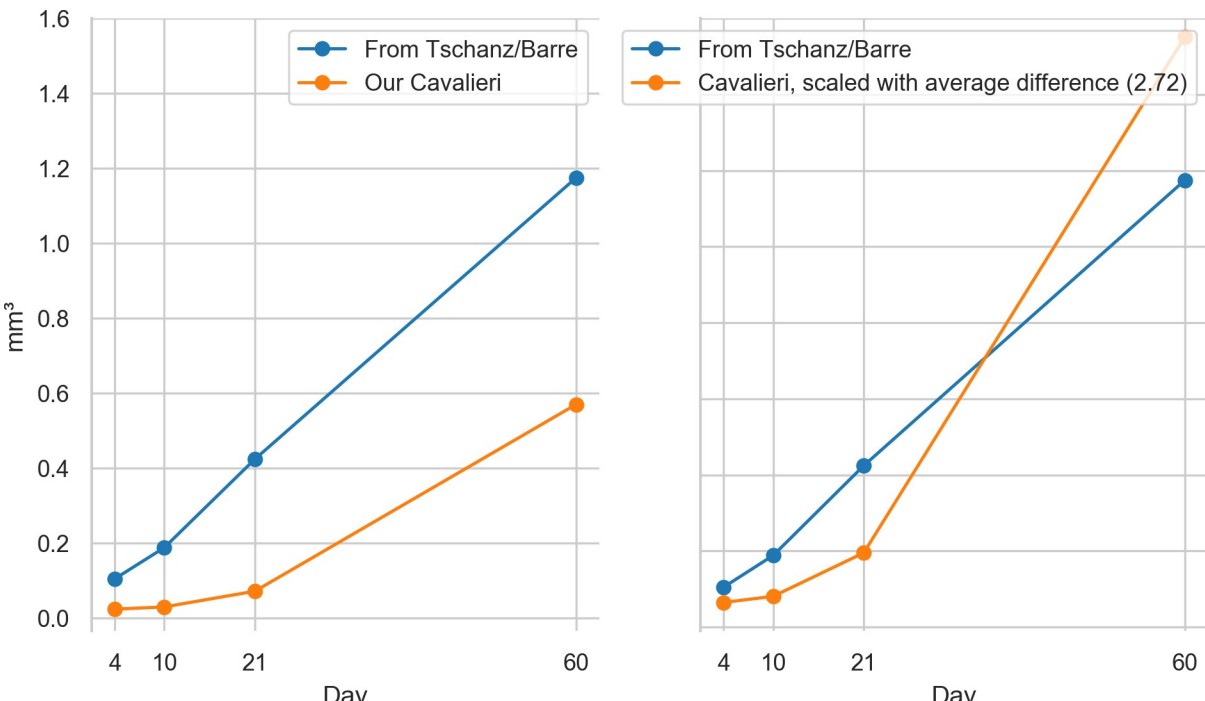

**Fig 6. Plot of the mean acinar volumes.** By dividing the parenchymal volumes of each lung estimated by Tschanz et al. (Table 1, [6]) by the mean number of acini estimated by Barré et al. [11] we can estimate the mean acinar volume for each measured time point (blue plot). We assessed the volumes of the acini by point counting (Cavalieri estimation, orange plot). Left: original data. Right: Our data, scaled by a linear correction factor. Our volumes of the acini are on average 2.07 times lower but show a similar increase over the studied period. While the estimation based on Tschanz et al. is from the entire right middle lobe and on Barré et al. globally for the whole organ, our estimation is solely based on the stereological analysis of the acini in the lower medial tip of the right lower lung lobe.

alveoli per acinus volume at days 10 and 21 compared to days 4 and 60 (all p-values are smaller than 5.4e-5, which is the p-value found between day 4 and day 21). Kreyling et al. [16] report maximal gold nanoparticle retention at this time point (see their Fig 1).

Tschanz et al. [6] stereologically estimated the mean total acinar volume as the sum of ductal and alveolar air spaces. The mean volume of the individual alveoli was found to be the smallest around postnatal day 21 (see their Fig 4). If we take the number of alveoli as one possible measure of acinar complexity, we postulate a high acinar complexity at days 10 and 21 while early and late in the lung development, the acinar complexity is relatively low. This also correlates with structural changes. In rats, the alveolar air spaces are large during the first two weeks of life, small at day P21 and medium afterward [6].

## Number of acini

Barré et al. have shown that the number of acini (e.g. counted acinar entrances) remains constant during postnatal lung development from day 4 to 60 [10], with a mean of 5840 ± 547 acini per lung (in another strain of animals, the data from the whole lungs is available for days 4 and 60). Even though we did not aim to assess the number of acini in this manuscript, calculating this number helps to validate our assessment and helps us to collate the data that can be derived from our stereological analysis to prior results.

Based on the acinar and parenchymal volume we calculated the number of acini and were able to verify the previously reported results to lie within our margin of error at days 4 and 60 (Barré et al.: 6236 ± 497 acini and 5612 ± 547 acini, respectively). As shown in Fig 7, our

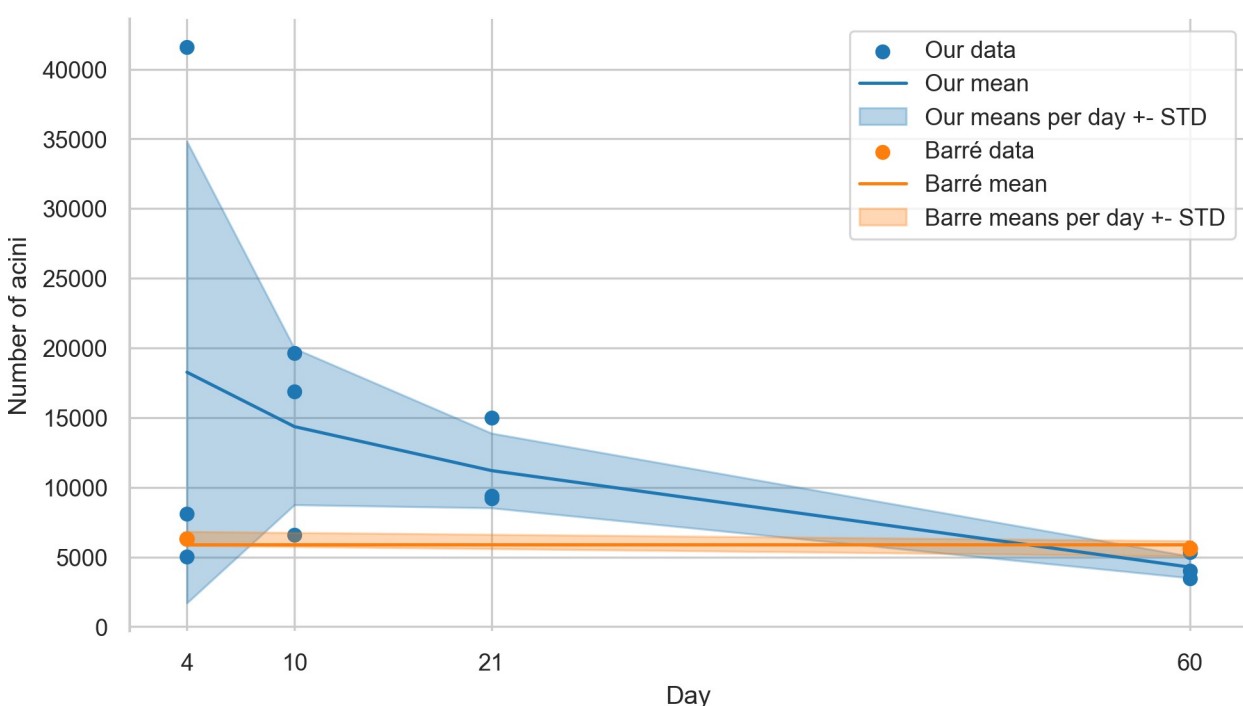

**Fig 7. Plot of the estimated number of acini.** The blue plot shows our estimation, based on the division of the parenchymal volume by the average volume of the acini. The orange plot is based on the data shown by Barré et al. [10, 11].

values ± standard deviation are overlapping the previously found values, even though we generally see a trend that the number of acini seems to decrease during lung development. This might be due to fusing of several acini to one or simply because we missed acini when performing the analysis of the large datasets.

## Number of alveoli per lung

In this study, we assessed the number of alveoli per acinus. By multiplication of the number of acini by the number of alveoli we estimate a total amount of alveoli per lung of 0.64 +- 0.34, 1.23 +- 0.39, 2.65 +- 0.57 and 3.01 +- 0.58 million alveoli for days 4, 10, 21, & 60, respectively. Tschanz et al. estimated these total numbers of alveoli per lung as 0.823 +- 0.01, 3.538 +- 1.26, 14.303 +- 3.08 and 19.297 +- 3.06 million alveoli for days 4, 10, 21, & 60, respectively [6], Table 1). On average, our results are five times smaller but show the same trend when linearly scaled with a factor of 5.04. The values from Tschanz et al. overlap well with our linearly scaled values, as shown in Fig 8.

As shown in a study by Osmanagic et al. [53] the total alveolar number shows a large variability between different labs. E.g. the reported total number of alveoli in adult C57BL6 mice differs by a factor of approximately 8. All the studies mentioned by Osmanagic et al. were performed in labs well-known for their quality of stereology-based investigations of lung morphology. Therefore, any simple bias or methodical error may not account for the reported differences. It seems to be that the total number of alveoli is dependent on unknown small methodical differences and minute differences in how the operator actually does count. The seemingly minuscule volume fraction of lung studied (around 1% or less) in this manuscript is —according to the stereological guidelines [30]—ample volume to gain mathematically correct results if the stereology is carried out correctly as we have done to the best of our knowledge.

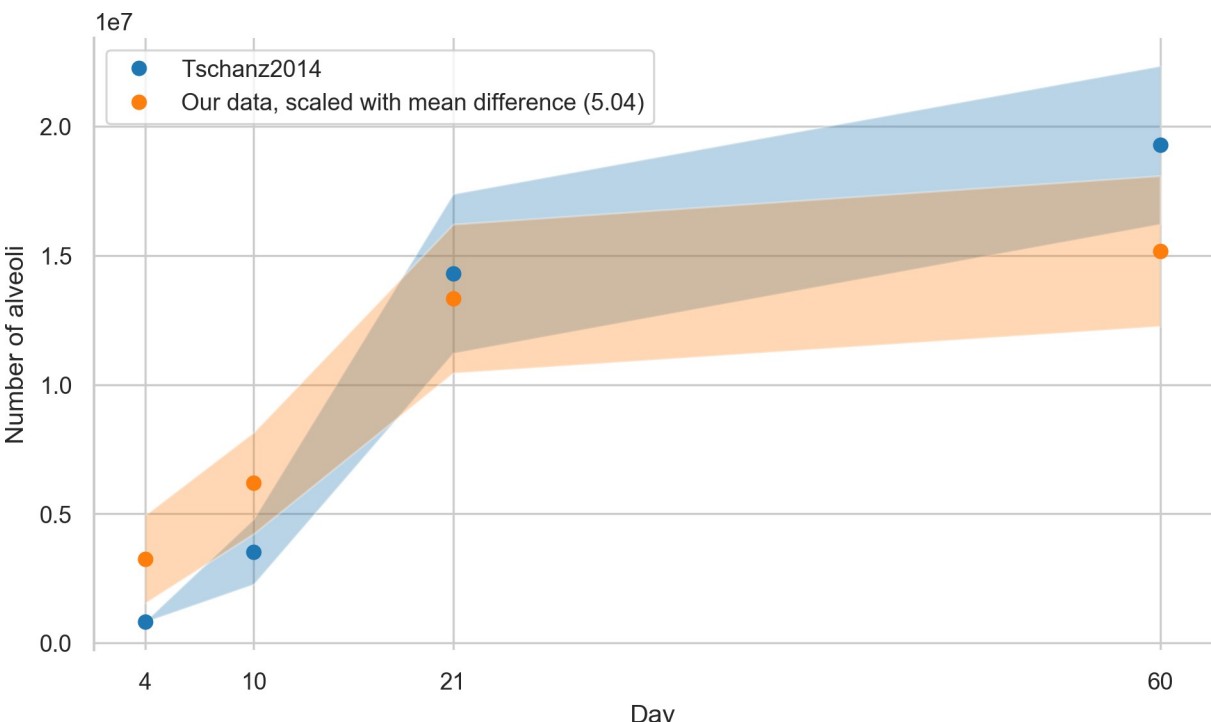

**Fig 8. Number of alveoli based on data from Tschanz et al. [6] together with our linearly scaled data (our data multiplied by 5.04).** Values are plotted with a band of the width of their standard deviation.

For human lungs for example a so-called sampling cascade is used to study details down to the thickness of the air-blood barrier (0.5 μm, see Fig 1 of [54] for a visualization of this cascade). The studied volume is in that case multiple decimal powers smaller than what is studied here while still being mathematically sound and accepted.

However, relative numbers as reported in this study are more reliable. This is especially the case when the comparison is done in groups of animals processed in parallel and counted by the same operator. We assessed the same lungs as Tschanz et al. [6], but performed our assessment based on a different imaging method and with two different persons performing the stereological assessment at two different points in time. Furthermore, we used the number of acini as determined by Barré et al. [11] which represents a dataset obtained by a third team and which introduces the effect of error propagation. This may explain the relative factor of 5.04 that we found as a difference. Even if it was shown that the right lower lobe represents a valid sample for the entire lung [10, 46], we cannot rule out a contribution to the factor by regional differences in the lung which we cannot account for in the present study.

## Volume of individual alveoli

By dividing the alveolar air space by the number of alveoli (both from Table 1 of [6]) we can estimate a mean alveolar volume of $5.91 \times 10^5$ μm$^3$ for day 4, of $2.49 \times 10^5$ μm$^3$ for day 10, of $1.34 \times 10^5$ μm$^3$ for day 21, and $2.93 \times 10^5$ μm$^3$ for day 60. On average, our values (calculated by dividing the mean acinar volume by the mean count of alveoli per acinus) are 1.81 times larger than these estimates of individual alveolar volume (our values day 4: $4.98 \times 10^5$ μm$^3$, day 10: $3.34 \times 10^5$ μm$^3$, day 21 $3.07 \times 10^5$ μm$^3$ and day 60 $8.12 \times 10^5$ μm$^3$), but scale equally during postnatal lung development. Tschanz et al. estimated the total alveolar volume without the

alveolar ducts. Since we directly estimated the volume fraction of the alveoli per acinus our number is actually expected to be larger.

## Additional characteristics of the acinus

We are well aware that there are considerably more parameters of the acinar characteristics which can be extracted from the acinar datasets we segmented for this study. Especially of interest are the microarchitecture in the acinus like the branching pattern of the intra-acinar airways or surface area characteristics of the acinar and alveolar walls. For this study we focused on the easily stereologically available parameters. Ongoing work in our group is aiming to describe additional parameters of the acinus microarchitecture (e.g. fractal dimension, surface curvature, etc.) but the present study did not aim at characterizing these parameters.

## Physiological relevance of the acinar size

It has been predicted by computational fluid dynamics simulations and by to-scale experiments that the amount and location of massless particle deposition, as well as ventilation, are depending on the size of the acini [12, 14, 55]. The predictions could be confirmed at least in one case, where mice received fluorescent particles during mechanical ventilation. In 3D-visualizations the particles were predominately detected in the proximal half of the acini [56].

Based on the above-mentioned knowledge and the data of this study, we propose the following. In the smallest acini we observed at postnatal day 4 acinar flows are mainly characterized by radial streamlines [49, 57]. Diffusion governs massless particle deposition [58]. As the acini increase in size, convective transport becomes more and more dominant in the proximal regions of the acini. In larger acini, the proximal region shows much higher particle retention than the distal one, because in this region the larger air-flow induces a circular current, which facilitates alveolar particle deposition. The proximal regions may even be viewed analogously to a filter capturing the particles whereas the distal regions do not receive significant amounts [12, 14, 15, 50, 51, 55].

Multi-breath gas-washout is also dependent on the size of the acini. Based on computational simulations of nitrogen multi-breath gas-washout it has been shown that small acini washout faster at the beginning and slower at the end of the washout as compared to larger acini [13]. Not surprisingly, the distribution of the acinar sizes also has an influence on the gas-washout. Furthermore, a clustering versus an even distribution of the locations of similar acinar sizes also affects gas-washout. Therefore, the location of an acinus of which size in the lungs has a significant influence on pulmonary ventilation and particle deposition.

The larger distribution of the acinar sizes at day 4 as compared to day 60 let us speculate what this may mean biologically. Physiologically, a larger distribution implies less homogeneous ventilation and an increased gas-washout time. Therefore, the decrease of the inhomogeneity of the acinar sizes may be understood as optimization of the gas-exchange region in the lungs of adult animals.

The consideration of the location is not only a theoretical one, because it has been shown in adult mice that the acini close to the pleura are larger than the central ones [22]. Therefore, acini of different size are not evenly distributed in the lungs.

## Conclusion

Newly formed acini show a large size distribution which is reduced by a factor of 7 until adulthood. Most likely the latter leads to more homogeneous ventilation of the lung and an increase of gas-washout which is equal to a decreased in the number of breaths needed for gas-washout.

To our best knowledge, no in-depth characterization of the acinar properties as presented here—i.e. spanning the lung development period—was available up to now. Even though our results are not in precise accordance with data calculated from previous studies, they match well when adjusted linearly.

The data presented in this study can contribute to improve computational simulations of pulmonary ventilation and particle deposition.

## Acknowledgments

We thank Mohammed Ouanella and Bettina de Breuyn for expert technical assistance. Federica Marone and Bernd Pinzer were of paramount importance with their support at the TOM-CAT beamline. Fluri Wieland was of great help with the statistical analysis.

We thank manubot [59] for providing us with the means to collaboratively write and craft this manuscript.

## Author Contributions

**Conceptualization:** Johannes C. Schittny.

**Data curation:** David Haberthür, Eveline Yao, Sébastien F. Barré, Johannes C. Schittny.

**Formal analysis:** David Haberthür, Tiziana P. Cremona, Stefan A. Tschanz.

**Funding acquisition:** Johannes C. Schittny.

**Investigation:** David Haberthür, Eveline Yao.

**Methodology:** David Haberthür, Johannes C. Schittny.

**Project administration:** David Haberthür, Johannes C. Schittny.

**Software:** David Haberthür.

**Supervision:** Johannes C. Schittny.

**Validation:** David Haberthür, Johannes C. Schittny.

**Visualization:** David Haberthür.

**Writing – original draft:** David Haberthür.

**Writing – review & editing:** David Haberthür, Eveline Yao, Sébastien F. Barré, Tiziana P. Cremona, Stefan A. Tschanz, Johannes C. Schittny.

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
