## [Decision Letter · Decision Letter 0]

2 Jun 2021

PONE-D-21-11732

Pulmonary acini exhibit complex changes during postnatal rat lung development

PLOS ONE

Dear Dr. Schittny,

Thank you for submitting your manuscript to PLOS ONE. After careful consideration, we feel that it has merit but does not fully meet PLOS ONE’s publication criteria as it currently stands. Therefore, we invite you to submit a revised version of the manuscript that addresses the points raised during the review process.

In particular, the aim of the manuscript could be stated more specifically, instead of expressing it in general terms, so that readers can know the importance of the work. Furthermore, citations are not fully appropriate and should be improved prior to publication.

We look forward to receiving your revised manuscript.

Kind regards,

Josué Sznitman

Academic Editor

PLOS ONE

Journal Requirements:

2. Please remove your figures from within your manuscript file, leaving only the individual TIFF/EPS image files, uploaded separately.  These will be automatically included in the reviewers’ PDF.

Reviewers' comments:

Reviewer's Responses to Questions

**Comments to the Author**

1. Is the manuscript technically sound, and do the data support the conclusions?

Reviewer #1: Yes

Reviewer #2: Yes

2. Has the statistical analysis been performed appropriately and rigorously? 

Reviewer #1: Yes

Reviewer #2: Yes

3. Have the authors made all data underlying the findings in their manuscript fully available?

Reviewer #1: Yes

Reviewer #2: Yes

4. Is the manuscript presented in an intelligible fashion and written in standard English?

Reviewer #1: Yes

Reviewer #2: Yes

5. Review Comments to the Author

Reviewer #1: Currently, this is the main laboratory, which has the expertise to perform the type of detailed morphological work described in the manuscript. In that regard, I really appreciate this project which is aimed to study the changes of detailed acinar morphology (the volume and the number of alveoli) throughout rat lung development. I also agree with the author’s choice of method, using synchrotron-imaging approach. I have the following criticism and comments on their stated aims (“Aims of this study”).

In the first paragraph of “Aims of this study”, the authors mentioned an aim related acinar structure.

Acinar structure

Due to the lack of precise and complete three-dimensional data, simulations of airflow in the lung are based on relatively simple acinar models. The question of how well these models represent lung physiology remains open until the necessary data and physiologically correct models are available. To contribute to the validation of the computational fluid dynamics simulations, we extracted individual acini and determined their volume and their number of alveoli throughout rat lung development.

While I generally agree with the authors’ stated aim, I wonder how the knowledge of detailed acinar structure ‒ opposite of global knowledge ‒ would change the predictions of the current computational fluid dynamics simulations of airflow qualitatively. I am not convinced that CFD simulations based on the current new data would yield new knowledge beyond what we already know experimentally [see ref. 15].

In the second paragraph of the “Aims of this study”, the authors mentioned the effects of individual variations of acinar volumes during the postnatal lung development on CFD.

Individual acinar volumes

While the global volume of the acini is well known and studied, only little is known about the range of the volume of individual acini throughout lung development. We developed the hereby presented method to analyze large amounts of acini over the course of postnatal lung development in a time-efficient manner since no data on large amounts of acini is available up to now. Our aim was to understand the size distribution of the acini during lung development including the contribution of the alveoli. The observed range decreased during postnatal lung development by a factor of 6–7, which represents a completely unexpected result. The obtained data are essential for further investigations, like the influence of the size of the acini on ventilation or gas-washout, respectively. E.g., we are currently using our computational fluid dynamics model to simulate exactly this influence (own unpublished results and [13]).

I agree with the authors that changes in the individual variation of acinar volumes would likely to change acinar fluid mechanics. It would be important, however, to illustrate the potential effect of the observed new data (i.e., a reduction in variation by a factor of 6-7) on change in fluid mechanics by briefly summarizing Ref. 13 and perhaps the authors’ unpublished results.

Please give us a rational that a voxel size of 1.48mm is sufficient to achieve the objective of the current study.

To make the references more appropriate, the authors should mention

Das GK, Anderson DS, Wallis CD, Carratt SA, Kennedy IM, Van Winkle LS. Novel multi-functional europium-doped gadolinium oxide nanoparticle aerosols facilitate the study of deposition in the developing rat lung, Nanoscale, 8:11518–11530, 2016

together with ref. 16 and they should mention that both of these publications agree with the original results of Ref. 15.

Reviewer #2: The authors here present their findings on murine lung development over a time course of 4 days to 60 days postnatally. By applying the tomographic X-ray imaging and semi-automatic quantification of lung acinar structure like acinar number, acinar volume, the number of alveoli per acinus, alveoli per volume, etc., the authors explicitly recorded the characterization of lung acinar development with several findings e.g., the acinar volume in early time points shows a large variations with a factor of 27 between largest and smallest acini and in adulthood it gets more homogenous with a factor of 4. The manuscript presents an important information for the lung community despite there are several discrepancies in such as the acini number and alveoli number compared with previous literature. The comments are following.

1. The authors described the text repeatedly especially for results and discussion part. The author could short the paper in a more concise way.

2. A better single acinus image is to be expected in Figure 1 as it is not intuitively clear how the authors obtained acinus delineation. For example, show a high resolution acinar image with full boundaries.

3. Why the authors displayed the figures in both e.g., linear scale and log. scale for the same results? They could consider using a scale “break” in y-axis to better illuminate their results.

4. The authors described the statistical P value with “better” many times, for example, figure 2, “all p values better than 1.9e-5, which is the one between days4 and 10”. Guess here the author meant “better” refers to “larger”. The latter (or other more specifically description) is more scientifically sound.

5. One of the important findings here is the decreased numbers of acini from day 4 to day 60. However, the results from the authors’ early study demonstrated a constant acinar number in rats, despite using a different strain of rats. The difference is so obvious and why this variation presented in old and current studies. Which result is more scientifically or physiologically correct and why?

6. Figure 6, 7, and 8 could be mentioned/placed in the results section rather than discussion part.

7. Checking out the misspellings and typos. Page 23, last paragraph. “All the studies mentioned my Osmanagic et al. were performed….”. “my” should be corrected to “by”?

Page 25, the last sixth line, “…acinar similar sizes effects gas-washout”. Here affects?

8. Page 24, last line. “our values day 4: 4.98 × 105 μm3, day 10:…”. “our values” is confusing here because the authors mentioned our values early with different numbers.

9. Discussion part, it is not clear that why the larger variation of acini size increases the gas-washout time.

6. PLOS authors have the option to publish the peer review history of their article (what does this mean?). If published, this will include your full peer review and any attached files.

Reviewer #1: No

Reviewer #2: No

---

## [Author Response · Author response to Decision Letter 0]

19 Jul 2021

The 'point-by-point' response to the reviewers has been uploaded as a PDF document.

---

## [Decision Letter · Decision Letter 1]

31 Aug 2021

Pulmonary acini exhibit complex changes during postnatal rat lung development

PONE-D-21-11732R1

Dear Dr. Schittny,

We’re pleased to inform you that your manuscript has been judged scientifically suitable for publication and will be formally accepted for publication once it meets all outstanding technical requirements.

Kind regards,

Josué Sznitman

Academic Editor

PLOS ONE

Additional Editor Comments (optional):

Reviewers' comments:

Reviewer's Responses to Questions

**Comments to the Author**

1. If the authors have adequately addressed your comments raised in a previous round of review and you feel that this manuscript is now acceptable for publication, you may indicate that here to bypass the “Comments to the Author” section, enter your conflict of interest statement in the “Confidential to Editor” section, and submit your "Accept" recommendation.

Reviewer #1: All comments have been addressed

Reviewer #2: All comments have been addressed

2. Is the manuscript technically sound, and do the data support the conclusions?

Reviewer #1: Yes

Reviewer #2: Yes

3. Has the statistical analysis been performed appropriately and rigorously? 

Reviewer #1: Yes

Reviewer #2: Yes

4. Have the authors made all data underlying the findings in their manuscript fully available?

Reviewer #1: Yes

Reviewer #2: Yes

5. Is the manuscript presented in an intelligible fashion and written in standard English?

Reviewer #1: Yes

Reviewer #2: Yes

6. Review Comments to the Author

Reviewer #1: All comments have been addressed. ……………………………………………………………………………………………………………………………………………………………………………………………

Reviewer #2: Thanks for the corrections made by the authors, which addressed all of my concern.

I would recommand to accept the current manuscript.

Indicate y-axis with break is not a misleading way of presenting data with a large range. Truncated bar graph, however, is considered a misleading method for y-axis display, as indicated from the link provided by the authors (en.wikipedia.org/wiki/Misleading graph#Truncated graph).

7. PLOS authors have the option to publish the peer review history of their article (what does this mean?). If published, this will include your full peer review and any attached files.

Reviewer #1: No

Reviewer #2: No

---

## [Editor Report · Acceptance letter]

29 Oct 2021

PONE-D-21-11732R1 

Pulmonary acini exhibit complex changes during postnatal rat lung development 

Dear Dr. Schittny:

I'm pleased to inform you that your manuscript has been deemed suitable for publication in PLOS ONE. Congratulations! Your manuscript is now with our production department. 

Kind regards, 

on behalf of

Prof. Josué Sznitman 

Academic Editor

PLOS ONE